

# Bayesian spatiotemporal modelling of wildfire occurrences and sizes for projections under climate change

## A step-by-step marked point process approach using INLA-SPDE

COMPUTO

ISSN 2824-7795

Juliette Legrand[1]    Biostatistics and Spatial Processes, INRAE, Avignon, France
François Pimont    Ecologie des Forêts Méditerranéennes (URFM), INRAE, Avignon, France
Jean-Luc Dupuy    Ecologie des Forêts Méditerranéennes (URFM), INRAE, Avignon, France
Thomas Opitz    Biostatistics and Spatial Processes, INRAE, Avignon, France

Date published: 2023-11-18    Last modified:

**Abstract**

Appropriate spatiotemporal modelling of wildfire activity is crucial for its prediction and risk management. Here, we focus on wildfire risk in the Aquitaine region in the Southwest of France and its projection under climate change. We study whether wildfire risk could further increase under climate change in this specific region, which does not lie in the historical core area of wildfires in Southeastern France, corresponding to the Southeast. For this purpose, we consider a marked spatiotemporal point process, a flexible model for occurrences and magnitudes of such environmental risks, where the magnitudes are defined as the burnt areas. The model is first calibrated using 14 years of past observation data of wildfire occurrences and weather variables, and then applied for projection of climate-change impacts using simulations of numerical climate models until 2100 as new inputs. We work within the framework of a spatiotemporal Bayesian hierarchical model, and we present the workflow of its implementation for a large dataset at daily resolution for 8km-pixels using the INLA-SPDE approach. The assessment of the posterior distributions shows a satisfactory fit of the model for the observation period. We stochastically simulate projections of future wildfire activity by combining climate model output with posterior simulations of model parameters. Depending on climate models, spline-smoothed projections indicate low to moderate increase of wildfire activity under climate change. The increase is weaker than in the historical core area, which we attribute to different weather conditions (oceanic versus Mediterranean). Besides providing a relevant case study of environmental risk modelling, this paper is also intended to provide a full workflow for implementing the Bayesian estimation of marked log-Gaussian Cox processes using the R-INLA package of the R statistical software.

*Keywords:* Climate projection, Integrated Nested Laplace Approximation, Marked log-Gaussian Cox process, Spatiotemporal model, SPDE approach, Wildfire

# Contents

[1]Corresponding author: juliette.legrand@univ-brest.fr

# 1  Introduction

As recently experienced in France during summer 2022, wildfires constitute a major threat for the environment and the society. Climate change leads to an increase in societal and economic risks related to wildfires (Riviere et al. 2022), and moreover to an increase in fire-prone regions worldwide (Abatzoglou, Williams, and Barbero 2019). In France, the Southeast historical region (where wildfire occurrence data have been systematically collected in the "Prométhée" database since the 1970s) has received a lot of attention in recent years due to the large amount of burnt areas (e.g. Pimont et al. 2021; Opitz, Bonneu, and Gabriel 2020). In this study, we consider another French region that draws attention to its wildfire activity and its potential increase under climate change, the Aquitaine region in Southwest France. To better understand the processes driving wildfire activity in this specific region, in particular its sensitivity to weather conditions, and to measure its potential evolution under climate-change, we seek to stochastically model and describe wildfire activities. Statistical modelling of wildfire activity is challenging because occurrence intensities and sizes of wildfires (corresponding to the area burnt by a wildfire) vary in space and time according to meteorological conditions, land cover and human activities, and these predictors may act differently on the probabilities of the ignition of fires (occurrence) and their propagation after ignition has taken place (size).

More specifically, each wildfire can be characterized in space and time, for instance by its location $s$ and time $t$ of ignition where $\{s \in \mathscr{S}, t \in \mathscr{T}\}$ would correspond to the space-time study domain. Doing so, wildfire occurrences can be seen as the realization of a spatiotemporal point process, a stochastic model for the occurrences of space-time events. Moreover, each point of this spatiotemporal point process can be associated with a numerical information, such as the burnt area of the corresponding fire. Adding this random quantitative mark, the spatiotemporal pattern of wildfire occurrences and

sizes can be viewed as a marked spatiotemporal point process. That is, there exists a random measure $N$ that counts the number of points in Borel sets $B \subset \mathcal{S} \times \mathcal{T}$ with intensity function $\Lambda(s, t)$ determining the expected number of points in any set $B$, i.e.

$$\mathbb{E}\left[N(B) \mid \{\Lambda(s, t)\}_{s \in \mathcal{S}, t \in \mathcal{T}}\right] = \int_B \Lambda(s, t) d(s, t).$$

In this study, we will consider such marked spatiotemporal point processes where the occurrences of wildfires are the points of the process and the burnt area the associated marks. But note that other quantitative marks have been considered in the wildfire risk literature, such as the duration of the wildfire (e.g. Quinlan, Díaz-Avalos, and Mena 2021).

In addition, we will assume that $N$ conditionally on $\Lambda$ is a Poisson point process with intensity function $\Lambda(s, t)$, meaning that

$$N(B) \mid \{\Lambda(s, t)\}_{s \in \mathcal{S}, t \in \mathcal{T}} \sim \text{Poisson}\left(\int_B \Lambda(s, t) d(s, t)\right)$$

and for any disjoint Borel sets $B_1, B_2 \subset \mathcal{S} \times \mathcal{T}$, $N(B_1)$ and $N(B_2)$ are independent conditionally on $\{\Lambda(s, t)\}_{s \in \mathcal{S}, t \in \mathcal{T}}$.

Within a Bayesian modelling framework, it then appears natural to consider log-Gaussian Cox processes (LGCPs, Møller, Syversveen, and Waagepetersen 1998) which are a particular case of Poisson point processes where $\{\log \Lambda(s, t)\}_{s \in \mathcal{S}, t \in \mathcal{T}}$ is assumed to be a Gaussian random field. A Bayesian interpretation of the LGCP is that we consider a log-Gaussian prior process for the intensity function of a Poisson process. LGCPs allow flexible inclusion of covariate information and are useful when points tend to occur in clusters, a behavior that is captured by the stochastic nature of the point process intensity function given by a log-Gaussian process. LGCPs have found numerous applications in risk modelling, especially for wildfires, but also for ecological data (e.g. Illian, Sørbye, and Rue 2012; Illian et al. 2013; Soriano-Redondo et al. 2019).

A fast, accurate and widely used Bayesian inference scheme when dealing with LGCPs is the integrated nested Laplace approximation framework (INLA, Rue, Martino, and Chopin 2009; Illian, Sørbye, and Rue 2012), which astutely exploits Laplace approximations (Tierney and Kadane 1986) and has proven to be computationally faster than simulation-based methods such as Markov chain Monte Carlo (MCMC) (e.g. Taylor and Diggle 2014). INLA allows estimating Bayesian hierarchical models and assumes that the latent structure of the model is a Gaussian Markov random field (GMRF), which means that the precision matrix (i.e., the inverse of the variance-covariance matrix) is sparse and therefore allows for fast computations even with high-dimensional Gaussian vectors with up to tens of thousands of latent variables. The stochastic partial differential equation (SPDE) approach is then often combined with INLA (providing the so-called INLA-SPDE approach) in order to approximate Gaussian random fields with Matérn covariance by GMRFs with sparse precision matrix (Lindgren, Rue, and Lindström 2011).

The INLA-SPDE approach has been intensively applied to wildfires. For instance, Pereira et al. (2013) have considered the occurrences of wildfires over Portugal taking into account specific topographic and land cover covariates, along with the average precipitation prior to the fire season. Serra et al. (2014) modeled wildfire occurrences in Catalonia given the potential causes of wildfires by considering a zero-inflated Poisson model. They also took into account topographic variables, the distance to anthropic areas and land uses. Another example is the work of Opitz, Bonneu, and Gabriel (2020), who modeled wildfire occurrences in the Southeastern core area in France by including predictors related to temperature and precipitation, as well as numerous covariates based on land use and land cover.

Table 1: First lines of the wildfire dataset.

| PIX | XL2e | YL2e | DEP | YEAR | DOY | FWI | FA | NB0.1 | BA |
|---|---|---|---|---|---|---|---|---|---|
| 6373 | 476000 | 2081000 | 24 | 2019 | 227 | 2.7619048 | 2831.982 | 0 | 0 |
| 6373 | 476000 | 2081000 | 24 | 2019 | 225 | 0.8491777 | 2831.982 | 0 | 0 |
| 6373 | 476000 | 2081000 | 24 | 2013 | 158 | 6.7485680 | 2831.982 | 0 | 0 |
| 6373 | 476000 | 2081000 | 24 | 2013 | 155 | 3.4603725 | 2831.982 | 0 | 0 |
| 6373 | 476000 | 2081000 | 24 | 2019 | 229 | 8.2332123 | 2831.982 | 0 | 0 |
| 6373 | 476000 | 2081000 | 24 | 2013 | 154 | 1.8662924 | 2831.982 | 0 | 0 |

In the following, we build on the previous study of Pimont et al. (2021) but we implement several extensions of their *Firelihood* model, a marked spatiotemporal log-Gaussian Cox process model for wildfire activities. Their model was initially applied in the Southeast of France where climatic conditions are sensibly different from the Aquitaine region. Note that to better account for burnt areas of extreme wildfires, Koh et al. (2023) extended the *Firelihood* model by considering a two-component mixture model for moderate and extreme wildfire sizes and other improvements that we include in our model. Specifically, we will focus on using the Gamma distribution for wildfire sizes, which was shown to provide a good fit in the historical Southeast region in Koh et al. (2023) without the need to construct rather complex mixture models. In Section 2, we describe the data used in this study. Section 3 presents the structure of our model and its inference using the INLA-SPDE approach. In particular, we detail a subsampling approach for occurrence observations that are zero to keep fully Bayesian inference with INLA-SPDE feasible. Then in Section 4, we generate wildfire activities over the historical period in order to assess the validity of our methodology. Finally, in Section 5, we derive future wildfire activities over the Aquitaine region under different climate scenarios.

## 2 Wildfire data

The wildfire data considered in this study are partially provided by the BDIFF database which collects information on wildfires since 2006 on the whole French territory but for which no stochastic models similar to Firelihood have been developed so far. Since BDIFF data are known to show some gaps, they have been completed with those from a public-private network working on territorial risk management (GIP ATGeRi). We extracted data from 2006 to 2020 in the former administrative Aquitaine region, corresponding to four French "départements" Dordogne, Gironde, Landes and Lot-et-Garonne (see the left display in Figure 1). Due to meteorological factors and specific human activities (agriculture, tourism), winter and summer wildfires correspond to two different fire regimes. Therefore, in this study, we focus on the summer wildfires, which are more numerous and on average also much bigger in terms of size, and we keep only wildfires that have occurred between May to October.

The meteorological data used are weather reanalysis data for variables such as temperature, precipitation, humidity and wind speed, provided by Météo France from the SAFRAN model (Vidal et al. 2010). These data are then combined in order to obtain a Fire Weather Index (hereinafter FWI), a unit-less indicator of fire danger. Since the SAFRAN model is defined on a regular grid of 8km resolution, wildfire data are aggregated to the SAFRAN pixels; i.e., for each pixel-day $B$, we calculate the number of wildfire occurrences in $B$; however, we do not aggregate the marks (i.e., the burnt areas) but continue working with the observed burnt area for each of the wildfire occurrences.

In the final dataset, we have for each pixel-day the corresponding daily FWI, the forest area FA (i.e. the fuel surface), the number of fires that are greater than 0.1 hectares NB0.1 (for measurements issues) and the burnt area BA in hectares (see Table 1).

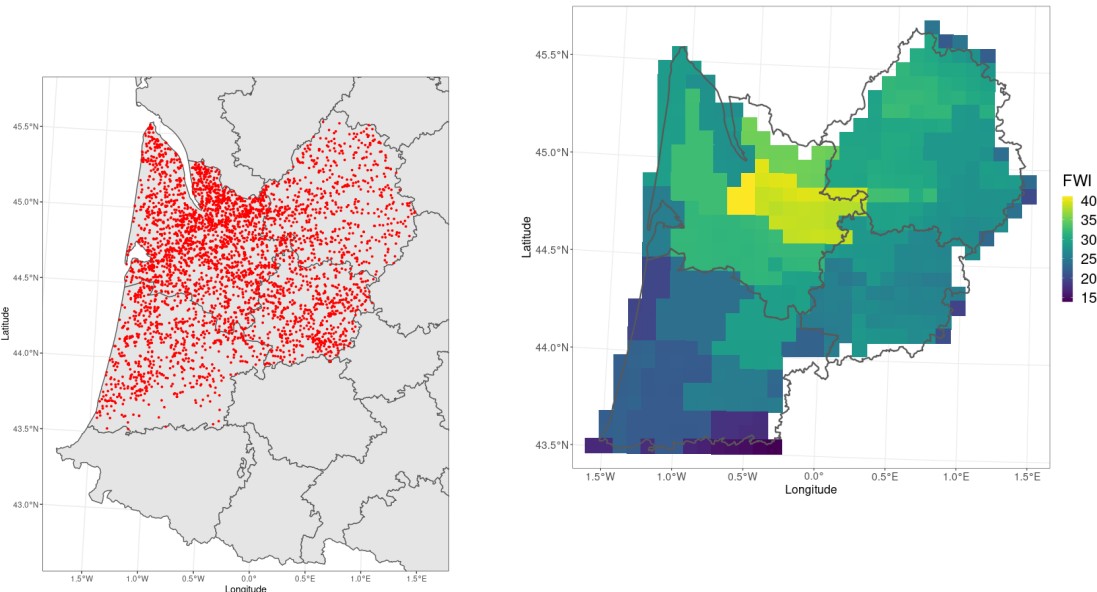

(a) Locations of wildfire activities during 2006-2020 over the region of interest.

(b) Spatial footprint of FWI observed on July 4, 2011

Figure 1: Maps of wildfire activity during the observation period and spatial variability of the Fire Weather Index (FWI), which is commonly used to aggregate the meteorological fire drivers into a single variable, for a given day.

## 3  Model and inference

### 3.1  Subsampling

There are 1308930 observations in the wildfire dataset, but only 2331 actual wildfires. In order to keep the model manageable for INLA, we subsample observations with count 0. This approach is similar to Koh et al. (2023) and it does not bias the analysis since we reweight the Poisson intensities estimated in the model to take into account the factor by which we have subsampled. For instance, if we keep on average one in ten zeros when fitting the model, we attribute a ten times larger Poisson intensity to the zeros remaining in the observations used for estimation; this works since Poisson intensity parameters are additive under convolution of Poisson-distributed random variables.

We keep all observations with positive count (corresponding to at least one actual wildfire) since they are the most informative and account only for a small number of observations. For such observations, we put a weight equal to 1.

Then, among the $n$ observations with 0 count, we only keep a number equal to $n_{ss}$ and much smaller than $n$, such that the associated weights are then equal to $n/n_{ss}$. However, since wildfires tend to occur most often for large FWI, we further use a stratified approach by sampling with a larger probability weight $p$ (i.e., the probability of keeping an observation) observations associated to large FWI values such that the model will be able to appropriately discriminate between wildfire presence (positive count) and absence (zero count) for relatively large FWI values. A large value of FWI is defined as exceeding a given high threshold $u_q$ given by a specific $q$-quantile, $q \in (0, 1)$. If we denote by $(w_k)_{1 \le k \le n_{ss}}$ the subsampling weights of observation with count 0, then, for observations with FWI greater than $u_q$, necessarily $n\mathbb{P}(\text{FWI} \ge u_q) = n_{ss}pw_k$, i.e. $w_k = n(1-q)/(n_{ss}p)$. Similarly, for FWI smaller than $u_q$, $n\mathbb{P}(\text{FWI} < u_q) = n_{ss}(1-p)w_k$ leading to $w_k = nq/(n_{ss}(1-p))$.

In the following, we choose $n_{ss} = 20000$, $q = 0.9$ and $p = 0.5$ (i.e. we keep as many observations with large FWI value as with lower FWI value) leading to a data set with dimension manageable by INLA on a personal computer and with computation times around the order of several minutes. Note that $p$ and $q$ are parameters that can be freely chosen based on the particularities of the data.

### 3.2  Model definition

In the following, the occurrences and wildfire sizes are considered as realizations of a spatiotemporal marked point process as explained in Section 1.

To account for the fact that the meteorological covariate FWI considered in this study is provided on a regular grid (see Section 2), we discretize our space-time study domain $\mathcal{S} \times \mathcal{T}$ and assume that the intensity of the LGCP $\Lambda(\boldsymbol{s}, t)$ is constant in each space-time cell $C_i \in \{1, \dots, M\}$, i.e.

$$\Lambda(\boldsymbol{s}, t) \equiv \Lambda_i \text{ if } (\boldsymbol{s}, t) \in C_i.$$

We then consider the occurrences. We model the number of fires, denoted $N_i$, in each space-time cell $C_i \in \{1, \dots, M\}$, corresponding to a SAFRAN pixel and a given day of the year. With these assumptions, the data discretized to the pixel grid are still coherent with the LGCP model introduced before, and we represent each pixel by its center coordinate.

In our model, we assume that for each space-time cell $C_i$, the number of fires $N_i$ is distributed according to Poisson distribution with a log-link function and a random intensity parameter, described in the following hierarchical structure with the data layer (first line), the latent process layer (second line)

and the hyperparameter layer (third line):

$$N_i \mid \Lambda_i, \boldsymbol{\theta} \sim \text{Poisson}(w_i \Lambda_i)$$
$$\log \Lambda_i = \beta_0 + f_{\text{YEAR}}(\text{YEAR}_i) + f_{\text{DOY}}(\text{DOY}_i) + f_{\text{FWI}}(\text{FWI}_i) + f_{\text{FA}}(\text{FA}_i) + f_{\text{Spatial}}(A_i) \quad (1)$$
$$\boldsymbol{\theta} = \left( \boldsymbol{\theta}^{\text{YEAR,size}}, \boldsymbol{\theta}^{\text{DOY}}, \boldsymbol{\theta}^{\text{FWI}}, \boldsymbol{\theta}^{\text{FA}}, \boldsymbol{\theta}^{\text{Spatial}} \right) \sim \text{Hyperpriors}$$

The random 1-dimensional functional effects $f_{\{\text{FWI,DOY, FA}\}}$ are defined through quadratic splines, for which SPDE prior models are available, and $f_{\text{Spatial}}$ is a spatial field. Finally, to better capture the yearly variability in the data, we add an iid random effect $f_{\text{YEAR}}$.

We work in a Bayesian framework, that is we put Gaussian prior distributions on all the random effects. For instance, for the function representing the effect of FWI in the predictor, we set the following prior structure:

$$\begin{cases} f_{\text{FWI}}(\text{FWI}_i) = \sum_{k=1}^{n} \beta_k^{\text{FWI}} b_k^{\text{FWI}}(\text{FWI}_i) \\ \boldsymbol{\beta}^{\text{FWI}} = (\beta_1^{\text{FWI}}, \dots, \beta_n^{\text{FWI}})^T \sim \mathcal{N}(\mathbf{0}, \mathbf{Q}_{\text{FWI}}^{-1}) \end{cases}$$

where $(b_k^{\text{FWI}})_{k \in \{1,\dots,n\}}$ are spline basis functions. Denoting $\boldsymbol{B}^{\text{FWI}} = \left( b_k^{\text{FWI}}(\text{FWI}_i) \right)_{(1 \le k \le n, 1 \le i \le M)}$ the matrix containing the values of the spline basis function at the observed FWI values, and similarly for the other covariates, the linear predictor in Equation 1 can be rewritten as follows

$$\log \boldsymbol{\Lambda} = (\log \Lambda_1, \dots, \log \Lambda_M) = \mathbf{1} \beta_0 + \sum_{\text{eff} \in \text{Effects}} \boldsymbol{B}^{\text{eff}} \boldsymbol{\beta}^{\text{eff}} \quad (2)$$

where Effects contains all the effects considered in the model Equation 1, i.e. Effects = $\{\text{FWI, FA, YEAR, DOY, Spatial}\}$. In the decomposition of Equation 2, the second term on the right-hand side is decomposed for each effect by a product between the effect (i.e., the coefficients $\beta^{\text{eff}}$ to be estimated) and a projector matrix (i.e., the deterministic matrices $B^{\text{eff}}$ calculated from the spline basis and the covariate values).

To fit this hierarchical model, we use the INLA framework (Rue, Martino, and Chopin 2009), which leverages an astutely designed deterministic approximation of the posterior distributions, unlike simulation-based methods such as MCMC.

For each component except the yearly effect, a latent Gaussian random field prior is approximated using the stochastic partial differential equations (SPDE, Lindgren, Rue, and Lindström 2011) approach. This approach allows us to approximate a continuous random field by a discrete random field with a finite number of Gaussian variables used as priors for basis function coefficients, where interpolation across continuous space provided is the deterministic basis functions. Shortly, recall that a Gaussian random field $f(\boldsymbol{s})$ on $\mathbb{R}^d$ can be obtained as the solution to the following SPDE

$$\left( \kappa^2 - \Delta \right)^{\alpha/2} \tau f(\boldsymbol{s}) = W(\boldsymbol{s}), \quad \alpha = \nu + d/2, \quad \boldsymbol{s} \in \mathbb{R}^d \quad (3)$$

where $\Delta$ is the Laplacian operator and $W(\boldsymbol{s})$ is a standard Gaussian white noise process. Then the only stationary solution to Equation 3 is a Gaussian random field with Matérn covariance function

$$\text{Cov}\left( f(\mathbf{0}), f(\boldsymbol{s}) \right) = \sigma^2 2^{1-\nu} (\kappa \|\boldsymbol{s}\|)^\nu K_\nu(\kappa \|\boldsymbol{s}\|) / \Gamma(\nu)$$

with Euclidean distance $\|\cdot\|$, Gamma function $\Gamma$, modified Bessel function of the second kind $K_\nu$, and $\sigma, \nu > 0$. In practice, we often fix $\nu = 1$ (as we do here), and then approximate solutions of Equation 3, having a Markov structure, are obtained using a finite element method with a triangulation of the space if $d = 2$, or splines if $d = 1$. Such solutions are Gaussian Markov random fields with sparse

precision matrices, allowing for fast numerical calculations even in high dimension (in terms of the number of basis functions, equal to the dimension of the corresponding latent Gaussian vector).

We then model the associated size components $\left(S_{i,1}, \ldots, S_{i,N_i}\right)$ given that $N_i > 0$, considered as the marks of the point process, through a Gamma distribution. As for the occurrence model, we consider the SPDE approach for the physical predictors FA and FWI, and an iid random effect for the year. We also consider an iid random effect for each "département", which leads to the following model:

$$
\begin{aligned}
\left(S_{i,1}, \ldots, S_{i,N_i}\right) \mid \alpha_i, \boldsymbol{\theta}^{\text{size}}, (N_i > 0) &\overset{\text{iid}}{\sim} \text{Gamma}(\alpha_i, \phi) \\
\log \alpha_i = \beta_0 + f_{\text{YEAR}}^{\text{size}}(\text{YEAR}_i) + f_{\text{FWI}}^{\text{size}}(\text{FWI}_i) &+ f_{\text{FA}}^{\text{size}}(\text{FA}_i) + f_{\text{Spatial}}^{\text{size}}(\text{DEP}_i) \\
\boldsymbol{\theta}^{\text{size}} = \left(\boldsymbol{\theta}^{\text{YEAR,size}}, \boldsymbol{\theta}^{\text{FWI,size}}, \boldsymbol{\theta}^{\text{FA,size}}\right) &\sim \text{Hyperpriors}
\end{aligned}
\tag{4}
$$

Note that with INLA, the parametrization of the Gamma likelihood is given by $\mathbb{E}(S_{i,k}) = \alpha_i$ and $Var(S_{i,k}) = \alpha_i^2/\phi$, for all $k \in \{1, \ldots, N_i\}$ where $\phi$ is a hyperparameter included in the vector $\boldsymbol{\theta}^{\text{FWI,size}}$. The Gamma distribution behaves similarly to a Gaussian distribution when $\phi$ is large, whereas its tails become more and more heavy when $\phi$ approaches 0.

## 3.3 INLA settings

For the SPDE-based effects, namely FWI, FA, DOY and Spatial, we construct a Matérn SPDE model with penalized complexity (PC) priors for the hyperparameters of the Matérn field (Fuglstad et al. 2019). The construction of the SPDE is achieved using the function `inla.spde2.pcmatern()`. We also define the associated projector matrix $B^{\text{eff}}$ with `inla.spde.make.A()`, mapping the projections of the SPDE to the observation points. Then, the function `inla.spde.make.index()` is used to define the indexes of the latent variable for the SPDE model (i.e., an identifier that runs from 1 to the number of basis functions).

The 1-dimensional effects $f_{\text{FWI,FA,DOY}}$ in Equation 1 are defined through quadratic splines with 5 knots for FWI and DOY, and 6 knots for FA. For FWI, we want to extrapolate values as a constant in cases where new covariate values used for prediction are larger than the observed covariate values used for fitting the model, so we impose a Neumann upper-bound condition corresponding to a zero first derivative.

Regarding the spatial effect $f_{\text{Spatial}}$, the triangulation mesh depicted in Figure 2 has 1051 nodes. In order to avoid non-stationary effects of the SPDE solution near the boundaries, we define two regions with a lower density of triangulation nodes in the outer region. Based on previous studies (e.g. Pimont et al. 2021), the parameters for the PC priors for the SPDE model (corresponding to exponential prior distributions for the standard deviation and the range parameter) are set using the following conditions: we set a probability of 50% to have a standard deviation larger than 1 and a probability of 5% to have a spatial range smaller than 50km.

## 3.4 Estimation of the occurrence model

To perform the model estimation, we gather all the information in a stack, a data format used in the R-INLA package that is appropriate for INLA and contains the data, the projection matrices and the different effects. In the model, we only have one fixed effect, which is the intercept, and five random effects. For the SPDE-based effects we include the indices of their associated SPDE. Then, we define the model formula as in Equation 1.

We then fit the model calling `inla()` with a number of user-specific settings to control how Laplace approximations are carried out and which posterior quantities are calculated.

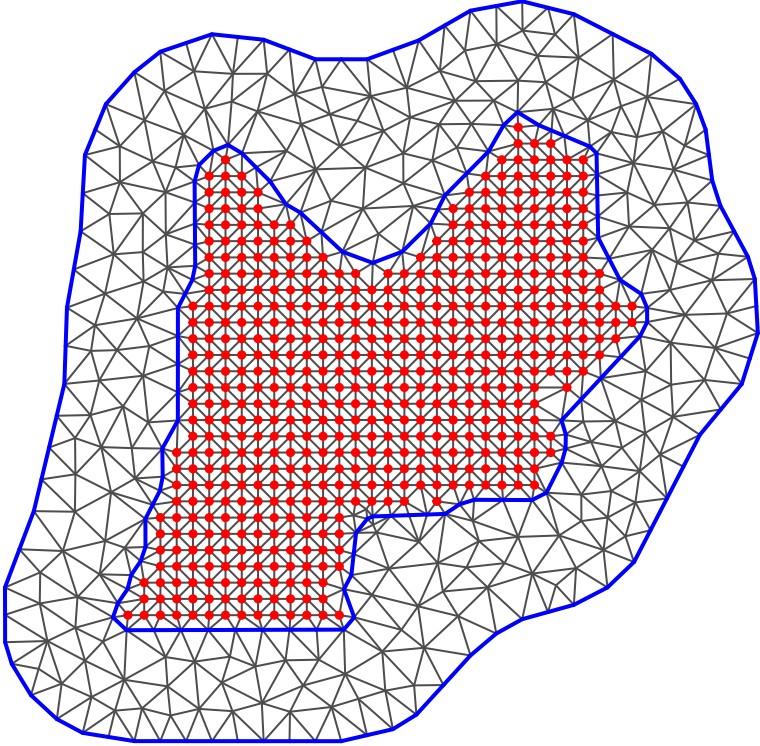

Figure 2: Constructed spatial mesh with 1051 nodes. Red dots represent the centers of the SAFRAN pixels for the study area.

We can visualize summaries of the posterior distributions of the random effects. For the SPDE effects, we have to project the effects on a 1-d (or 2-d for the spatial effect) grid that contains the initial mesh. The new projector matrix (i.e., the matrix $B$ containing the new values of covariates and evaluated spline bases) is obtained with the function `inla.mesh.projector()`. Results are depicted in Figure 3.

From Figure 3, we can conclude that the yearly effect captures an inter-annual variability that cannot be described by the physical parameters FWI and FA. Regarding the seasonal effect DOY, we see that it decreases in mid-September, after the high summer heat. Both FWI and Forest area (FA) effects increase almost linearly, which is something that we expect since wildfire activity increases with FWI and the amount of fuel material. Looking at the spatial effect, we can observe a high spatial correlation between the different locations and clear separated clusters highlighting different fire regimes. An interpretation of these results is that FWI and FA are able to explain a substantial part of the spatiotemporal variability in wildfire activity, but that there also remains strong spatiotemporal residual correlation that is captured by the other random effects. Lastly, the magnitude of the effects appears to be smaller for the yearly effect than for the other effects.

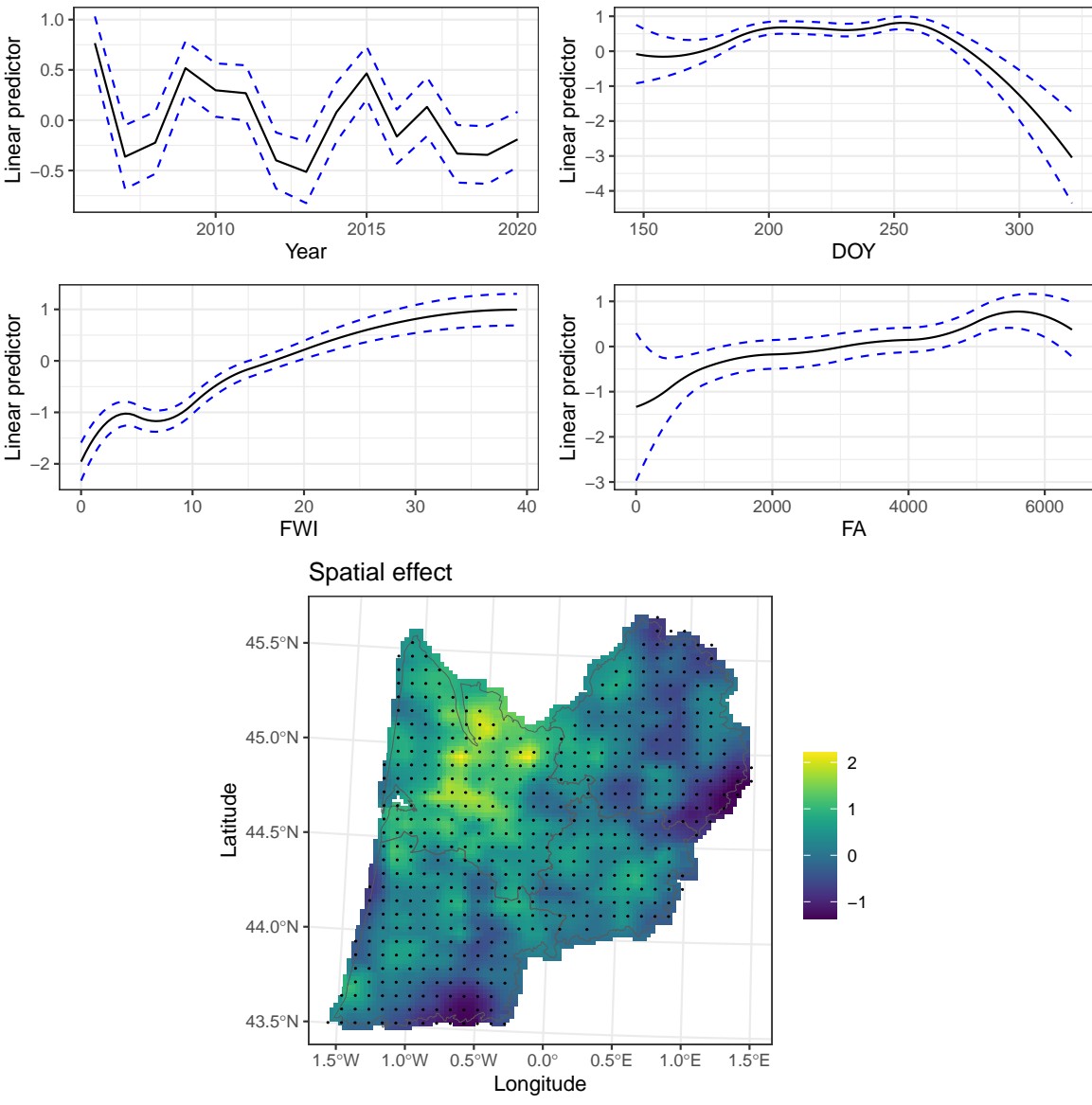

Figure 3: Partial effects of the occurrence model with the simulated data.

## 3.5 Estimation of the size model

Similarly to the occurrence model, we use INLA to perform the size model estimation. But prior to that, we need to build the new projection matrices for the SPDE effects defined in Equation 4 since we will consider hereafter only data such that the burnt area is greater than 0.1ha. We here perform the estimation of the size model separately from the estimation of the occurrence model, which is possible as long as we do not construct a model where some of the random effects are used in both the occurrence model and the size model, i.e., where there are shared random effects, such as in Koh et al. (2023). Separate estimation of the two models strongly reduces the overall computational cost for running INLA.

Again, after running the INLA estimation, we can plot the estimated random effects, see Figure 4. It can be seen that large wildfires tend to be associated with large values of FWI and FA. The spatial effect is almost null except for the north-east region for which we have a negative effect. Finally, looking at the amplitude of the effects, the FWI has the strongest influence on wildfire sizes.

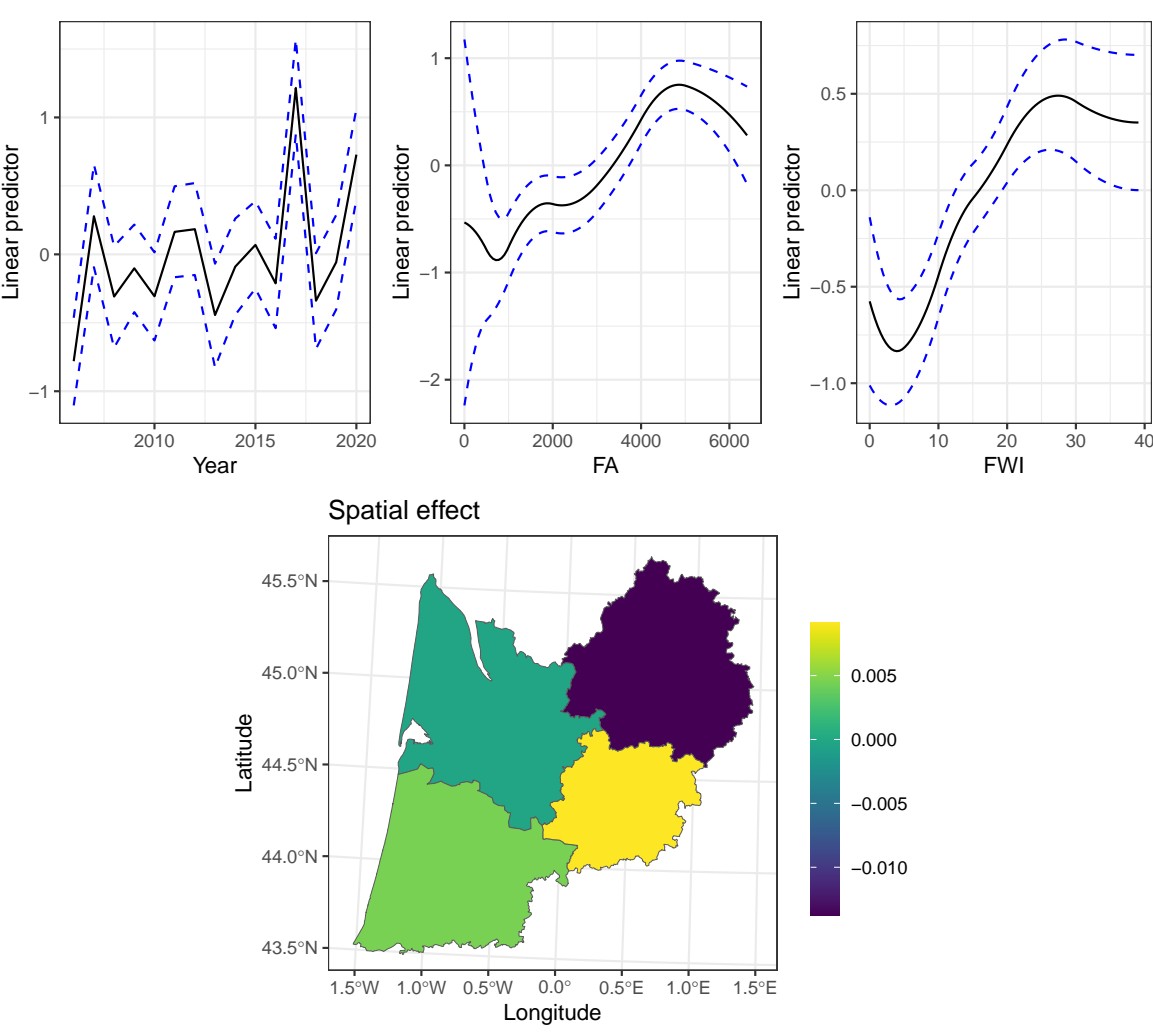

Figure 4: Partial effects of the size model.

# 4 Wildfire simulations from the posterior distribution for the observation period (2006-2020)

Before applying the fitted model to climate projections, we wish to assess the validity of our model. We here focus on the capacity of the model to reproduce the observed wildfires during the study period (2006-2020) with appropriate posterior uncertainty. R-INLA implements a method to obtain independent samples from the posterior distributions of hyperparameters and latent Gaussian variables, which can then be combined with new covariate data to calculate Monte-Carlo estimations of any posterior quantity of interest.

Note that the following results correspond to simulations obtained from the original data, which are not made available to reproduce the study. See Appendix B for more details.

## 4.1 Occurrence component

For the occurrence model, we first have to sample from the posterior distribution of all the coefficients in the occurrence model and then combine them with new effect values, using the additivity of Equation 2.

Hereinafter, we perform $n = 100$ simulations and results are depicted in Figure 5. The top line panels depict the spatial patterns of observed and simulated wildfire occurrences. In the bottom line, the yearly aggregated occurrences are shown on the left. Then, we looked for a specific year the daily and weekly aggregation (middle and right panels) of simulated wildfire occurrences. We chose to examine the year 2010, but any other year could have been considered.

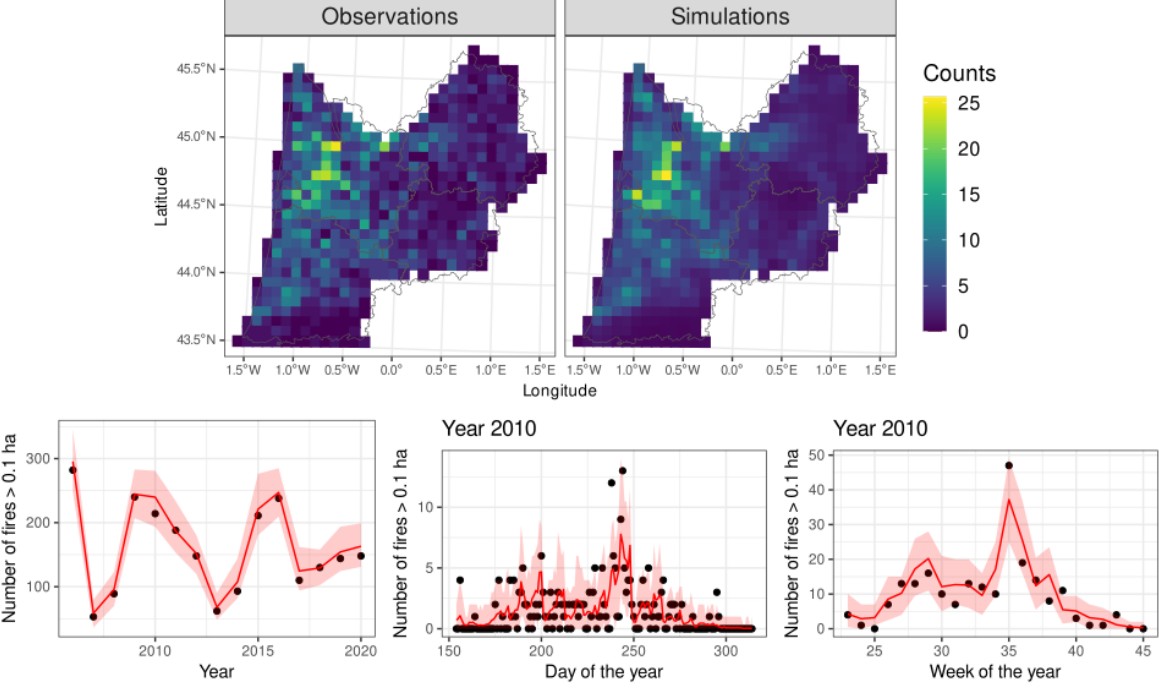

Figure 5: top: Spatial patterns of observed wildfire occurrences (left) and simulated occurrences (right). bottom: Simulated wildfire occurrences with 95% pointwise confidence intervals (in red) and observations (black dots).

Figure 5 highlights that the model successfully recovers both the spatial and temporal pattern of wildfire occurrences. Almost all observations are within the uncertainty bands of the posterior model.

The temporal aggregations also highlights the temporal trends and stochasticity of wildfire regimes, with for instance a stronger wildfire activity at the end of August.

## 4.2 Size component

As for the occurrence model, we illustrate the applicability of our model through simulations and compare the simulated sizes with the historical data (see Figure 6). The simulation scheme is as follows: for each of the 100 previously simulated samples of wildfire occurrences, we generate the associated sizes by sampling the posterior distributions of the fitted size model effects and use the additivity of the linear predictor as defined in Equation 4.

Simulations are presented in Figure 6. Again, the spatial aggregation is shown in the top line panels. The yearly aggregated burnt areas are depicted in the bottom-left panel. Finally, the middle and right panels in the bottom line depict the weekly aggregated occurrences of fires greater than 1 ha and 10 ha respectively.

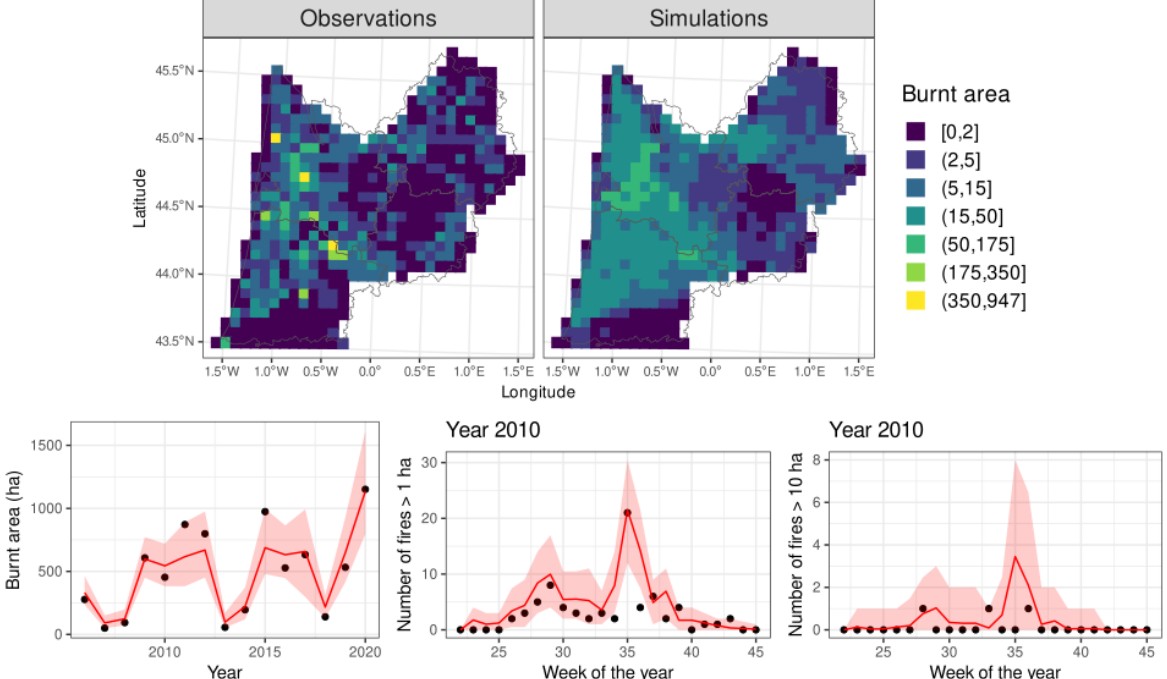

Figure 6: top: Spatial patterns of observed wildfire sizes (left) and simulated sizes (right). bottom: Simulated wildfire sizes with 95% pointwise confidence intervals (in red) and observations (black dots).

The size simulations depicted in Figure 6 show that the model successfully recovers the spatial pattern but misses some of the the most extreme values in terms of coverage of pointwise credible intervals. Unlike the wildfire occurrences, the distribution of burnt areas is heavy-tailed and more difficult to predict. Still, looking at the temporal aggregation the simulations provide satisfying results and almost all observations are within the uncertainty bands. We note that for the largest fires (i.e. greater than 10 ha), the sample size being much smaller, the uncertainties are larger.

## 5 Future wildfire simulations derived from climate model projections

We consider hereafter four different climate models under two climate scenarios RCP4.5 and RCP8.5 where the second one is more pessimistic in terms of expected global warming than the first one. The models, together with the institutes having developed them written in parentheses, are as follows: IPSL-CM5A (Institut Pierre-Simon Laplace, France), MPI-ESM (Max Planck Institut für Meteorologie, Germany), HadGEM2 (Met Office Hadley Center, UK) and CNRM (Météo-France, France).

For simulated FWI values in the climate change projections that fall outside the range of historical FWI values, we extrapolate the function $f_{\text{FWI}}$ as a constant. For the forest surface FA in each pixel, we extrapolate the historical values in a constant manner, that is, we use the values available for 2018.

We perform $n = 20$ realizations of pixel-day occurrences and generate for each occurrence its associated size. The occurrences and sizes can then be aggregated over various spatial and temporal scales to study the potential evolutions of future wildfire risk. In Figure 7, we depict the results at an annual scale. Interannual variability in simulated counts and sizes remains relatively high, even after averaging the twenty realizations over the whole study area. To smooth the projected curves and identify long-term trends in wildfire activity, we implemented a 1D-SPDE INLA model, given by Equation 5. The basis representation for the yearly effect is defined through a quadratic spline and we set PC priors for the prior function such that the probability to have a standard deviation larger than 100 is equal to 0.5 and we fix a range value of 30 years. This choice is motivated by the fact that a 30-year period is often used to calculate averages in climatological studies. Moreover, to assess if there is a general upward trend in the smoothed curve, we include the rescaled year as a linear covariate, i.e., as a fixed effect. The rescaling is constructed such as to interpret the coefficient $\beta_1$ as a decadal effect. Therefore, the role of the spline function $f_{\text{YEAR}}$ is to model nonlinear deviations from the general linear trend, $\beta_0 + \beta_1 (\text{YEAR}_i - 2020)/10$. For the sake of identifiability, we set Dirichlet boundary conditions for the spline function, such that it takes value zero at both boundaries.

$$
\begin{aligned}
Y_i \mid \mu_i, \theta^{\text{YEAR}} &\sim \text{Lognormal}(\mu_i, \sigma) \\
\mu_i &= \beta_0 + \beta_1 (\text{YEAR}_i - 2020)/10 + f_{\text{YEAR}}(\text{YEAR}_i) \\
\theta^{\text{YEAR}} &\sim \text{Hyperprior}
\end{aligned}
\tag{5}
$$

Since the posterior sampling from the fitted model takes a lot of memory to run, we have stored the results beforehand.

Looking at Figure 7, the four climate models provide substantially different results. The IPSL-CM5A and CNRM show no significant trend either for occurrences or for sizes under both scenarios. Among the two other models, MPI-ESM shows a clearer trend: by 2100, the number of wildfire occurrences can be expected to double on average under the most pessimistic emission scenario. The associated wildfire sizes are also increasing, going from 500 ha in 2020 to up to 1500 ha by the end of the century. This evolution can be better measured by looking at the posterior distribution of the decadal linear effect defined in Equation 5. Summary statistics for $\beta_1$ are reported in the Appendix A. Under the RCP8.5 scenario, the HadGEM-RCA4 and MPI-ESM models lead to significantly positive $\beta_1$ estimates since their 95% credible intervals do not contain zero.

To capture potential spatial variability of projected wildfire activities, we also considered the spatial aggregation of the simulated occurrences and sizes during the end of the projection period (2070–2100), results are depicted in Figure 8.

From Figure 8a, it can be seen that the spatial pattern of wildfire occurrences will remain essentially

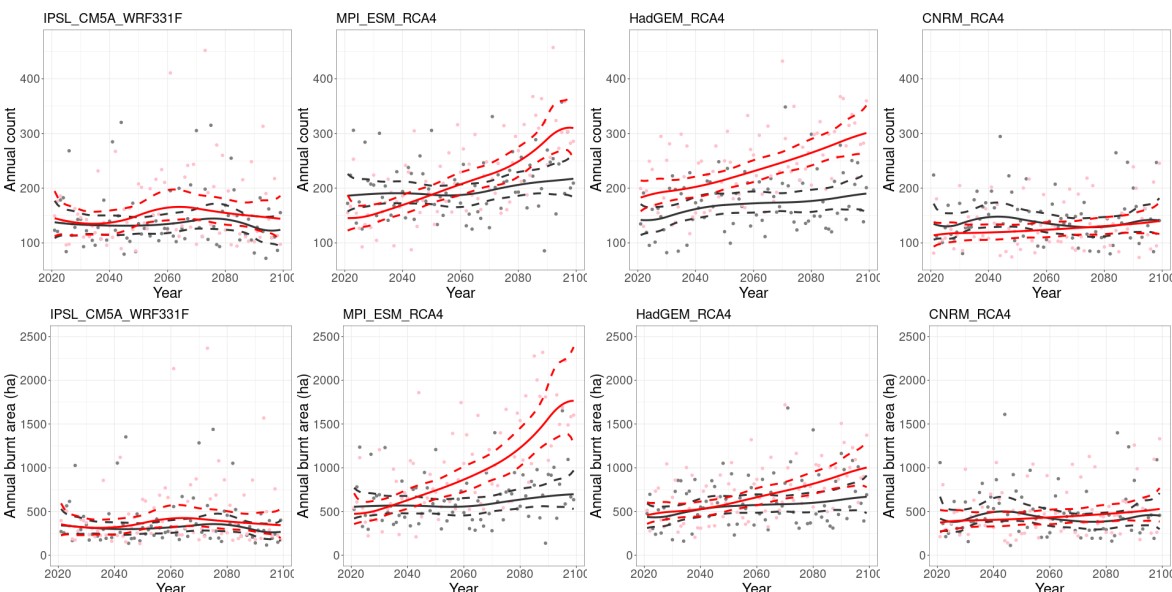

Figure 7: Annual means of wildfire occurrences (top-line) and sizes (bottom-line) for the four climate models and the two emission scenarios (RCP4.5 in black, RCP8.5 in red). Dots represent annual averages calculated over twenty samples from the full posterior model, continuous curves report the posterior mean fit of the INLA model used to smooth curves, and dashed lines indicate the corresponding 95% credible intervals.

the same, but the intensities within each pixel may change. Under the scenario RCP 4.5 most models show no strong significant increasing trend. However, under the pessimistic scenario RCP 8.5, MPI-ESM model seems to predict a substantial increase in wildfire activities. To a lesser extent, a similar observation can be made for models CNRM and HadGEM2.

Looking at the projected burnt areas Figure 8b, similarly to the simulations performed over the observation period in Section 4.2, predictions are quite noisy which we attribute to the fact that burnt areas are relatively heavy-tailed, such that a relatively small number of values can have a relatively strong influence on the calculation of the mean. But again, under the scenario RCP 8.5, MPI-ESM model shows a significant increase in the average annual wildfire size compared to the observation period.

These experiments seem to indicate that for the Aquitaine region the climate-related vulnerability of forests to wildfires could increase to a lesser extent in the future than in the historical core wildfire area in the Southeastern France.

## 6  Discussion

A first remark should be made concerning the most recent wildfire events: as the 2022 weather data are not yet available, the 2022 summer fire season was not considered in this study. Therefore, the projections obtained above do not take into account the relatively extreme wildfire activities with several very large fires that have been observed in the Western part of France during this period, and the resulting projections could potentially have been more pessimistic in terms of the future increase of wildfire risk in the study region. The results we obtain point towards an increase in future wildfire risk. However, the uncertainty about the future climate remains large and propagates through to projected wildfire risk. Indeed, the weather simulations of the four climate models considered here lead to clearly significant increases only in some cases for rather pessimistic scenarios of greenhouse

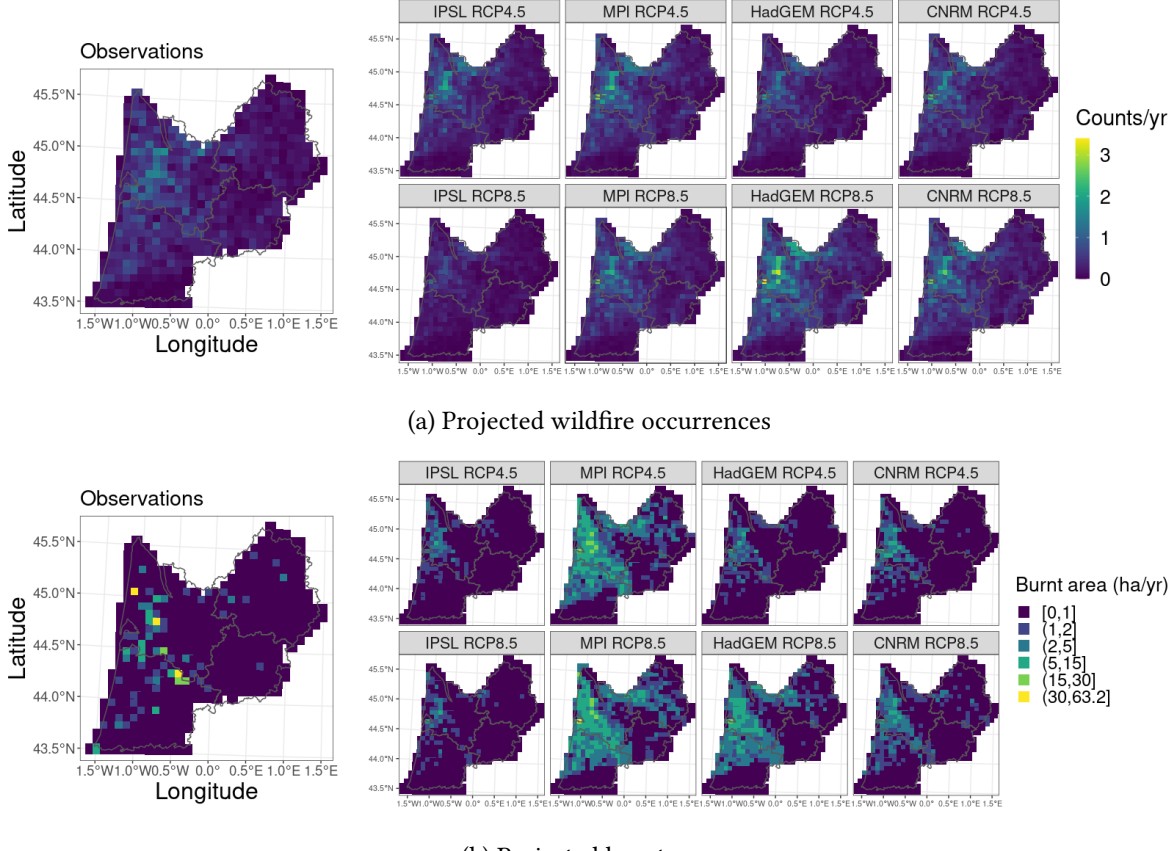

(a) Projected wildfire occurrences

(b) Projected burnt area

Figure 8: Spatial patterns of wildfire activity during the end of the projection period (2070−2100) for each climate models and associated climate scenarios. Values correspond to the mean values over the period and are compared to the mean values during the observation period (2006-2020) displayed on the left-hand side.

Table 2: Estimated slopes of the linear predictor defined in Eq. (5) for each climate model

| | Occurrence | | | Size | | |
|---|---|---|---|---|---|---|
| | mean | 0.025quant | 0.975quant | mean | 0.025quant | 0.975quant |
| IPSL_CM5A_WRF331F 4.5 | -0.013 | -0.073 | 0.039 | -0.029 | -0.132 | 0.061 |
| IPSL_CM5A_WRF331F 8.5 | -0.003 | -0.077 | 0.055 | -0.007 | -0.137 | 0.090 |
| MPI_ESM_RCA4 4.5 | 0.020 | -0.020 | 0.061 | 0.029 | -0.037 | 0.096 |
| MPI_ESM_RCA4 8.5 | 0.094 | 0.033 | 0.133 | 0.164 | 0.069 | 0.228 |
| HadGEM_RCA4 4.5 | 0.037 | -0.008 | 0.082 | 0.052 | -0.021 | 0.124 |
| HadGEM_RCA4 8.5 | 0.064 | 0.030 | 0.098 | 0.101 | 0.042 | 0.157 |
| CNRM_RCA4 4.5 | 0.003 | -0.053 | 0.058 | 0.003 | -0.105 | 0.100 |
| CNRM_RCA4 8.5 | 0.027 | -0.014 | 0.077 | 0.046 | -0.025 | 0.131 |

gas emissions.

This work presented a step-by-step methodology for the modelling of spatiotemporal marked point processes, that has been applied to the modelling of wildfire activities in the Southwest region of France. Due to the high stochasticity involved in wildfire activity but also in climate-change projections, and due to the complex processes and data that have to be modeled, Bayesian hierarchical modeling provides an appropriate framework for including various observed predictors and random effects into a model that allows for accurate predictions with precise uncertainty assessment. Our model includes additive random effects for various components of the linear predictors, such as nonlinear effects of continuous covariates, spatial random effects and temporal random effects. The SPDE approach provides flexible Gaussian prior distributions for such effects with two hyperparameters for the variance and the correlation range, and the INLA method allows for fast and reliable Bayesian inference even with complex and high-dimensional structures of the latent linear predictor and the likelihood model of the data.

In addition, we also presented how INLA can be used to smooth relatively noisy simulations of projected time series of risk occurrences, here based on combining posterior simulations of model parameters with new weather-related covariates obtained from climate model output. Our smoothing approach based on a Bayesian hierarchical model is an attractive statistical alternative to the classical filtering approaches from signal processing, since it can lead to more interpretable results while at the same time providing uncertainty envelopes.

We want to emphasize that our modeling approach for spatiotemporal marked point processes can also be used in other contexts. In ecology, for example, researchers are interested in modelling the distribution of species in space and time over a given study area: the occurrences of the spatio-temporal process could be the observation locations, and the marks could refer to certain characteristics (traits) of the observed individuals. In particular, we plan to construct INLA-SPDE models similar to the one presented here to project how species distributions evolve in response to present and future climatic change (see e.g. Guillot et al. 2022; Laxton et al. 2023).

# Appendix A

Estimates of $\beta_1$ as defined in Equation 5 are depicted in the following table, highlighting for which climate model there is a positive trend in wildfire activities (in red).

# Appendix B

The results depicted in Figures Figure 5 and Figure 6 are obtained with the original wildfire data. Since these data are confidential, the code presented in this paper should be applied to a simulated data set, obtained with the model developed above using the original data (see Supplementary materials).

In the following, we show the figures that should be obtained using the proposed simulated data set and the code developed in the previous sections.

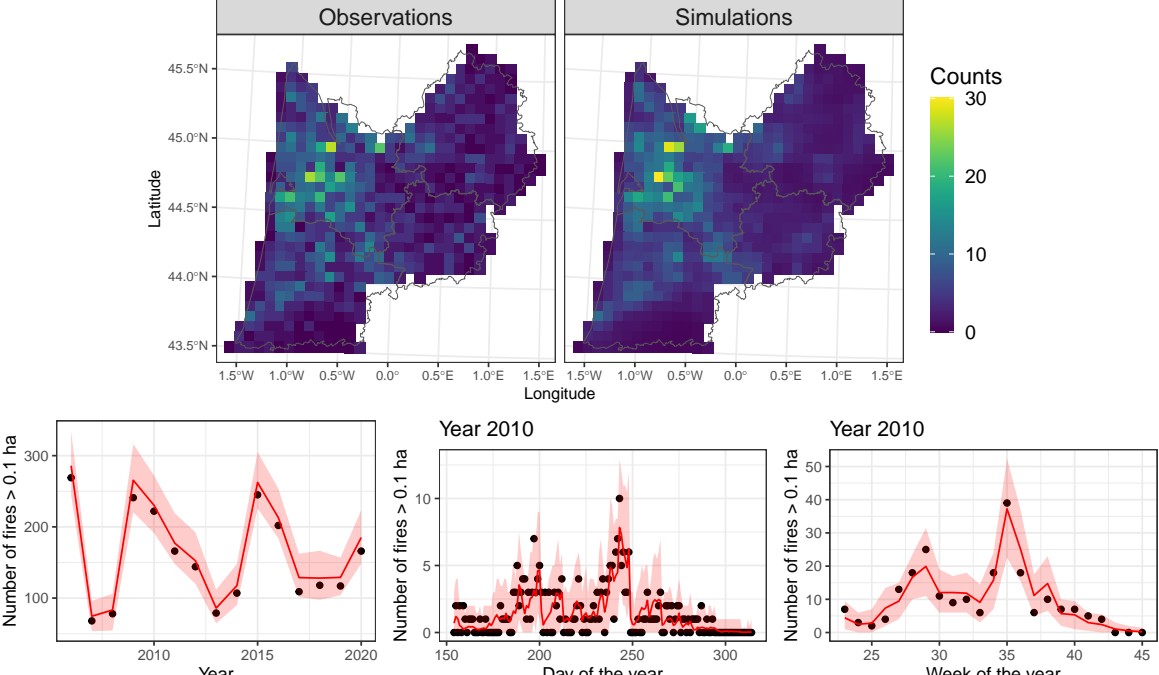

Figure 9: top: Spatial patterns of observed wildfire occurrences (left) and simulated occurrences (right). bottom: Simulated wildfire occurrences with 95% pointwise confidence intervals (in red) and observations (black dots).

# Ackowledgments

The authors are grateful to Météo-France for making the SAFRAN data available.

# Supplementary materials

All data used in this study are available at the following link: https://doi.org/10.5281/zenodo.7870592. Note that for confidentiality reasons, the wildfire data provided correspond to simulated data from the model developed in this work.

The R codes are available on GitHub: https://github.com/jlegrand35/wildfire_activities

# References

# Session information

R version 4.3.1 (2023-06-16)

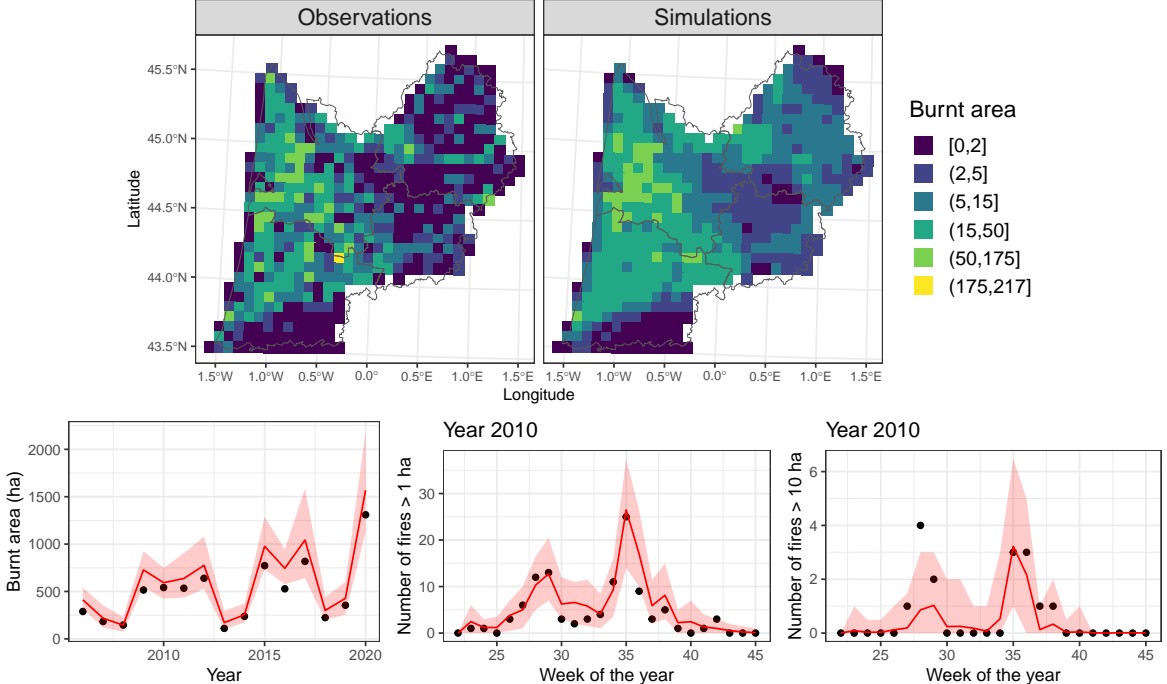

Figure 10: top: Spatial patterns of observed wildfire sizes (left) and simulated sizes (right). bottom: Simulated wildfire sizes with 95% pointwise confidence intervals (in red) and observations (black dots).

```
403  Platform: x86_64-pc-linux-gnu (64-bit)
404  Running under: Ubuntu 22.04.3 LTS
405
406  Matrix products: default
407  BLAS:    /usr/lib/x86_64-linux-gnu/openblas-pthread/libblas.so.3
408  LAPACK: /usr/lib/x86_64-linux-gnu/openblas-pthread/libopenblasp-r0.3.20.so;  LAPACK version 3.10.0
409
410  locale:
411   [1] LC_CTYPE=C.UTF-8       LC_NUMERIC=C           LC_TIME=C.UTF-8
412   [4] LC_COLLATE=C.UTF-8     LC_MONETARY=C.UTF-8    LC_MESSAGES=C.UTF-8
413   [7] LC_PAPER=C.UTF-8       LC_NAME=C              LC_ADDRESS=C
414  [10] LC_TELEPHONE=C         LC_MEASUREMENT=C.UTF-8 LC_IDENTIFICATION=C
415
416  time zone: UTC
417  tzcode source: system (glibc)
418
419  attached base packages:
420  [1] stats4    stats     graphics  grDevices datasets  utils     methods
421  [8] base
422
423  other attached packages:
424   [1] kableExtra_1.3.4  knitr_1.43        INLA_23.10.28      Matrix_1.6-1
425   [5] viridis_0.6.4     spdep_1.2-8       sf_1.0-14          spData_2.3.0
426   [9] dplyr_1.1.3       gridExtra_2.3     ggplot2_3.4.3      stringr_1.5.0
427  [13] evd_2.3-6.1       fields_15.2       viridisLite_0.4.2 spam_2.9-1
```

```
[17] splancs_2.01-44   sn_2.1.1            sp_2.1-1

loaded via a namespace (and not attached):
 [1] dotCall64_1.0-2    gtable_0.3.4       xfun_0.40
 [4] lattice_0.21-9     numDeriv_2016.8-1.1 vctrs_0.6.3
 [7] tools_4.3.1        generics_0.1.3     parallel_4.3.1
[10] tibble_3.2.1       proxy_0.4-27       fansi_1.0.4
[13] pkgconfig_2.0.3    KernSmooth_2.23-22 webshot_0.5.5
[16] lifecycle_1.0.3    farver_2.1.1       fmesher_0.1.2
[19] compiler_4.3.1     deldir_1.0-9       MatrixModels_0.5-2
[22] munsell_0.5.0      mnormt_2.1.1       htmltools_0.5.6
[25] maps_3.4.1         class_7.3-22       yaml_2.3.7
[28] pillar_1.9.0       classInt_0.4-10    wk_0.8.0
[31] boot_1.3-28.1      rvest_1.0.3        tidyselect_1.2.0
[34] digest_0.6.33      stringi_1.7.12     labeling_0.4.3
[37] splines_4.3.1      fastmap_1.1.1      grid_4.3.1
[40] colorspace_2.1-0   cli_3.6.1          magrittr_2.0.3
[43] utf8_1.2.3         e1071_1.7-13       withr_2.5.0
[46] scales_1.2.1       httr_1.4.7         rmarkdown_2.24
[49] evaluate_0.21      s2_1.1.4           rlang_1.1.1
[52] Rcpp_1.0.11        glue_1.6.2         DBI_1.1.3
[55] xml2_1.3.5         renv_1.0.2         svglite_2.1.1
[58] rstudioapi_0.15.0  jsonlite_1.8.7     R6_2.5.1
[61] systemfonts_1.0.4  units_0.8-4
```

Abatzoglou, John T., A. Park Williams, and Renaud Barbero. 2019. "Global Emergence of Anthropogenic Climate Change in Fire Weather Indices." *Geophysical Research Letters* 46 (1): 326–36. https://doi.org/https://doi.org/10.1029/2018GL080959.

Fuglstad, Geir-Arne, Daniel Simpson, Finn Lindgren, and Håvard Rue. 2019. "Constructing Priors That Penalize the Complexity of Gaussian Random Fields." *Journal of the American Statistical Association* 114 (525): 445–52. https://doi.org/10.1080/01621459.2017.1415907.

Guillot, Gilles, Ali Arab, Janine Bärbel Illian, and Stéphane Dray. 2022. "Editorial: Advances in Statistical Ecology: New Methods and Software." *Frontiers in Ecology and Evolution* 9. https://doi.org/10.3389/fevo.2021.828919.

Illian, Janine B., Sara Martino, Sigrunn H. Sørbye, Juan B. Gallego-Fernández, María Zunzunegui, M. Paz Esquivias, and Justin M. J. Travis. 2013. "Fitting Complex Ecological Point Process Models with Integrated Nested Laplace Approximation." *Methods in Ecology and Evolution* 4 (4): 305–15. https://doi.org/https://doi.org/10.1111/2041-210x.12017.

Illian, Janine B., Sigrunn H. Sørbye, and Håvard Rue. 2012. "A toolbox for fitting complex spatial point process models using integrated nested Laplace approximation (INLA)." *The Annals of Applied Statistics* 6 (4): 1499–1530. https://doi.org/10.1214/11-AOAS530.

Koh, Jonathan, François Pimont, Jean-Luc Dupuy, and Thomas Opitz. 2023. "Spatiotemporal wildfire modeling through point processes with moderate and extreme marks." *The Annals of Applied Statistics* 17 (1): 560–82. https://doi.org/10.1214/22-AOAS1642.

Laxton, Megan R., Óscar Rodríguez de Rivera, Andrea Soriano-Redondo, and Janine B. Illian. 2023. "Balancing Structural Complexity with Ecological Insight in Spatio-Temporal Species Distribution Models." *Methods in Ecology and Evolution* 14 (1): 162–72. https://doi.org/https://doi.org/10.1111/2041-210X.13957.

Lindgren, Finn, Håvard Rue, and Johan Lindström. 2011. "An Explicit Link Between Gaussian Fields and Gaussian Markov Random Fields: The Stochastic Partial Differential Equation Approach."

*Journal of the Royal Statistical Society: Series B (Statistical Methodology)* 73 (4): 423–98. https://doi.org/10.1111/j.1467-9868.2011.00777.x.

Møller, Jesper, Anne Randi Syversveen, and Rasmus Plenge Waagepetersen. 1998. "Log Gaussian Cox Processes." *Scandinavian Journal of Statistics* 25 (3): 451–82.

Opitz, Thomas, Florent Bonneu, and Edith Gabriel. 2020. "Point-Process Based Bayesian Modeling of Space–Time Structures of Forest Fire Occurrences in Mediterranean France." *Spatial Statistics* 40: 100429. https://doi.org/10.1016/j.spasta.2020.100429.

Pereira, Paula, Kamil Feridun Turkman, Maria Antónia Amaral Turkman, Ana Sá, and José MC Pereira. 2013. "Quantification of Annual Wildfire Risk; a Spatio-Temporal Point Process Approach." *Statistica* 73 (1): 55–68.

Pimont, François, Héléne Fargeon, Thomas Opitz, Julien Ruffault, Renaud Barbero, Nicolas Martin-StPaul, Eric Rigolot, Miguel Riviere, and Jean-Luc Dupuy. 2021. "Prediction of Regional Wildfire Activity in the Probabilistic Bayesian Framework of Firelihood." *Ecological Applications* 31 (5): e02316. https://doi.org/10.1002/eap.2316.

Quinlan, José J., Carlos Díaz-Avalos, and Ramsés H. Mena. 2021. "Modeling Wildfires via Marked Spatio-Temporal Poisson Processes." *Environmental and Ecological Statistics* 28 (3): 549–65. https://doi.org/10.1007/s10651-021-00497-1.

Riviere, M., F. Pimont, P. Delacote, S. Caurla, J. Ruffault, A. Lobianco, T. Opitz, and J. L. Dupuy. 2022. "A Bioeconomic Projection of Climate-Induced Wildfire Risk in the Forest Sector." *Earth's Future* 10 (4): e2021EF002433. https://doi.org/https://doi.org/10.1029/2021EF002433.

Rue, Håvard, Sara Martino, and Nicolas Chopin. 2009. "Approximate Bayesian Inference for Latent Gaussian Models by Using Integrated Nested Laplace Approximations." *Journal of the Royal Statistical Society: Series B (Statistical Methodology)* 71 (2): 319–92. https://doi.org/10.1111/j.1467-9868.2008.00700.x.

Serra, Laura, Marc Saez, Jorge Mateu, Diego Varga, Pablo Juan, Carlos Diaz-Ávalos, and Håvard Rue. 2014. "Spatio-Temporal Log-Gaussian Cox Processes for Modelling Wildfire Occurrence: The Case of Catalonia, 1994–2008." *Environmental and Ecological Statistics* 21: 531–63.

Soriano-Redondo, Andrea, Charlotte M. Jones-Todd, Stuart Bearhop, Geoff M. Hilton, Leigh Lock, Andrew Stanbury, Stephen C. Votier, and Janine B. Illian. 2019. "Understanding Species Distribution in Dynamic Populations: A New Approach Using Spatio-Temporal Point Process Models." *Ecography* 42 (6): 1092–102. https://doi.org/https://doi.org/10.1111/ecog.03771.

Taylor, Benjamin M., and Peter J. Diggle. 2014. "INLA or MCMC? A Tutorial and Comparative Evaluation for Spatial Prediction in Log-Gaussian Cox Processes." *Journal of Statistical Computation and Simulation* 84 (10): 2266–84. https://doi.org/10.1080/00949655.2013.788653.

Tierney, Luke, and Joseph B. Kadane. 1986. "Accurate Approximations for Posterior Moments and Marginal Densities." *Journal of the American Statistical Association* 81 (393): 82–86. https://doi.org/10.1080/01621459.1986.10478240.

Vidal, Jean-Philippe, Eric Martin, Laurent Franchistéguy, Martine Baillon, and Jean-Michel Soubeyroux. 2010. "A 50-Year High-Resolution Atmospheric Reanalysis over France with the Safran System." *International Journal of Climatology* 30 (11): 1627–44.

