# OpenReview forum: "Bayesian spatiotemporal modelling of wildfire occurrences and sizes for projections under climate change"
_Computo — Accepted by Computo_

### Review · Reviewer_hLqv · 2024-03-21

**Summary Of Contributions:**

The authors construct a marked space-time log-Gaussian cox process model for wildfire occurrences and burnt areas, which allows for covariates in components of the model that separately describe the distribution for occurrences and burnt area sizes. The authors provide a "how to" guide on fitting this model using the INLA Bayesian inference framework, with the SPDE approach used to approximate the latent spatial Gaussian random effects. This very flexible model is then fitted to data sampled over the Aquitaine region of France, and the fit is assessed via simulations from the model. Non-linear covariate effects (on both components of the model) are illustrated and interpreted, alongside spatial random effects. In the final section of the paper, the authors use various climate projections as input into their fitted model, in order to make inference about the future (projected) wildfire distribution.

**Audience:**

Yes

**Claims And Evidence:**

No

**Requested Changes:**

* Abstract: "...corresponding to the Southeast". I guess this should be South west?
* The description of the marked point process in Section 1 doesn't seem to describe the marks at all. From my understanding, the presented model is just the usual space-time point process.
* l55 "...it then appears natural to consider". Why is this the case?
* Table 1; What do the first four columns represent?
* Section 2. How are the data collected? Where does the FA variable come form? Why do the authors choose to keep only wildfires between May and October?
* The caption in Figure 1 appears unfinished.
* Section 3. I didn't quite understand how zero observations would occur for a marked point process dataset. I also didn't follow very well the exposition in the second paragraph.
* l146, "...p and q are parameters that can be freely chosen based on the data". Some discussion of this choice would be helpful. It seems quite an important modelling choice, particularly as the authors choose to take p = 0.5.
* Section 3.2. What value does M take in your application?
* When f_FWI is defined, $n$ is used for the number of spline basis functions, but has already been used for the sample size.
* Eq. (3). Details on $\kappa$ and $\tau$ are missing.
* Eq. (4). Why is the spatial effect here dependent on the departement, rather than the spatial location (as in (1))?
* The choice of a Gamma likelihood for the burnt areas seems odd. The Gamma distribution has exponential tails, meaning it does not have the "heavy tails" mentioned by the authors. This may be why the model fit for the wildfire sizes suffers in the tails. Koh et al (2023) use a generalised Pareto distribution, which seems to me a far better choice (and is already implemented in INLA). Did the authors consider this instead?
* Section 3.3. How were the number of knots chosen? What are PC priors and why are they set as described? How is the number of nodes in the spatial mesh chosen?
* Section 3.4. Do the authors have any explanation for the "bump" in the spline estimates for FWI (Figure 3)? Note as well that the caption for this figure is not self-contained, and it would be difficult to understand this figure without the accompanying text.
* Section 4 would be better placed before the results in Sections 3.4 and 3.5. It would be good to see evidence that the model fits well before interpreting its outputs.
 * Section 4.1. Why do you focus on 2010?
* I did not follow equation (5). The notation Y_i has not been used and there has been no mention of the lognormal distribution thus far (although this may be a more appropriate model than the gamma distribution, see above comment).

**Strengths And Weaknesses:**

The model used is very interesting and seems to fit the data well. However, it wasn't fully clear to me how the proposed model differs from the Firelihood model or the model proposed by Koh et al. (2023), both of which are referenced by the authors in Section 1. The authors should clarify the differences at the end of Section 1 to highlight the novelty of their approach.

The use of climate projections to infer future wildfire behaviour is an interesting idea. However, I have a couple of concerns about the approach. First of all, it doesn't seem like the authors make any attempt to de-bias the climate projections. The models are fitted using covariates derived from a reanalysis, and then the climate model data is fed into the model "off-the-shelf". Without any form of calibration of the different climate projections to the observations from the reanalysis, I can't say that I find the results in Section 5 particularly meaningful. It would make more sense to me if the reanalysis data was used to form the projections, potentially via a simple linear regression model, and then this information used in the analysis of Section 5 (rather than the projections from the four separate GCM models). Secondly, the authors set extrapolated f_FWI and f_FA values to a constant. It's not clear how often (if at all) this extrapolation occurs during the procedure described in Section 5 or what these constants are. I assume the constant is just taken to be the point estimate of the spline at the edge of the domain, e.g., at FWI = 40, you take the predictor equal to approx 1 for FWI > 40 (from Figure 3). However, I imagine there is quite a large amount of uncertainty in the estimates of the splines at the edge of the domain, which is a fairly common problem with spline-based models (there's evidence of overfitting in FA and FWI in Figure 4, where we see the predictor decrease with large FA or increase with low FWI). If inference in Section 5 for future wildfire distributions relies quite heavily on this choice of extrapolation constant, how are we able to trust the results? It's also not fully clear to me what happens with f_YEAR. Figures 4 and 5 show some very rough behaviour in estimates of f_YEAR; if Section 5 relies on extrapolation of these splines, then I find it very difficult to trust any results in Section 5.

The paper is presented as a "how to" guide for modelling marked log-Gaussian Cox processes with INLA, but there's a lot of key information, mostly related to the model formulation, which is missing. Seemingly arbitrary modelling choices are made throughout and not justified, and parts of the paper appear to be unfinished. I've provided specific details below, but I recommend that the authors carefully check the entirety of the paper before resubmitting.

The journal has a particular focus on reproducibility of studies. Given that the data used in the study are not publicly available, the study described is not reproducible. Following my reservations about Section 5 and overfitting in the spline estimates, I'd argue that the results of the study are not convincingly compelling as to use the SAFRAN data in place of data that are publicly available. However, with the additional clarity in the model procedure that I have requested below, I still think the paper would make a nice addition to Computo as a "how to" guide for modelling.

---

### Review · Reviewer_iH5f · 2024-03-28

**Summary Of Contributions:**

The manuscript entitled "Bayesian spatiotemporal modelling of wildfire occurences and sizes for projections under climate change'' proposes an interesting approach for modeling wildfire activity. The introduced framework relies on Bayesian spatiotemporal modeling and emphasis is made on the resolution of the model using the INLA-Stochastic Partial Differential Equation approach. The paper is well-written and the proposed modelling of wildfire is well introduced. The validation of the framework is performed on the observed wildfires in the Aquitaine during the period 2006-2020. A simulation study is furthermore conducted to evaluate climate model projection.

**Audience:**

Yes

**Broader Impact Concerns:**

No concern

**Claims And Evidence:**

Yes

**Requested Changes:**

## Major Compulsary Comments

My main concern is about the contribution of the submitted work. It is unclear to me whether the contribution is about the analysis of wildfires in the Aquitaine region or whether it is about the use of a bayesian spatiotemporal modelling.
- On the one hand, regarding the modelling approach, what is the novelty compared to the approach proposed by Koh et al. (2023)?
- Regarding the analysis in the Aquitaine region, it is hard to understand how the methodology is generalizable to other datasets (and /or other contexts). Several choices were made with respect to the applicative context but are not discussed in terms of modeling limitations. For example: In P6L243, are the choices of the parameters $n_{ss}$,$q$ and $p$ driven by the application or for a modeling purpose?
- The estimation of the occurence  and the size are performed independently. This point should be clarified and discussed from a modeling (or estimation) point-of-view and regarding the application (is this hypothesis relevant in the Aquitaine dataset?).

## Minor Comments


- Although the size of the grid (8km resolution) is forced by the use of the SAFRAN model, it would have been interesting to have a discussion on the impact of the choice of the resolution in space and also in time. I wonder how the results will be different for a different resolution : for example 32km in space and 1week in time.
- It could have been useful to detail how the FWI is calculated/computed and to provide a reference in P2L113.
- In section ``5. Future wildfire simulations ..."  the climate scenarios are not well detailed. It could be useful to have a table summarizing the main characteristics of each scenario in order to better appreciate the comparative study.
- In P8L195, why do the author consider a random effect for each ``département''?

## Typos

- P385: folowing -> following

**Strengths And Weaknesses:**

## Strengths
- The paper is well-written and easy to read.
- The INLA-SPDE pipeline is well-conducted.

## Weakness
- the innovation of the contribution is not clear.
- Modeling choices are not sufficiently motivated whether it is for the model itself or for its resolution

---

### Review · Reviewer_DqP1 · 2024-04-10

**Summary Of Contributions:**

The article is very well-written and well-organized. I thoroughly enjoyed reading it. The author provides a comprehensive description of spatiotemporal modeling of high-dimensional fire occurrences and sizes within the Bayesian inferential paradigm, utilizing INLA for inference. Fire occurrences are modeled using the Poisson response with a random intensity measure of log-Gaussian Cox types, while mark sizes are modeled using the gamma distribution with spatially varying means incorporating spatiotemporal structures. The author also suggests a posterior simulation-based approach for future projections under two commonly known climate scenarios. Overall, the flow of the manuscript is handled well, and the content is well-presented.

**Audience:**

Yes

**Claims And Evidence:**

Yes

**Requested Changes:**

## Main comments
 - Section 3.1: Do you have any insight into how these subsampling schemes affect spatial dependencies in your model? I agree that the use of the weights you introduced doesn't introduce bias, given how you've defined them, which is justified by the conditional independence assumption at the data level, making convolution of Poisson acceptable. It's a clever approach. However, I'm concerned that at the latent process level, subsampling might alter the original spatial dependence pattern in the zeros as well as the transition from zeros to non-zero occurrences. I understand that it's costly to account for all observed locations that indeed have large proportions of zeros. But it would be beneficial to provide a brief overview of how this subsampling of zeros might impacts spatial dependencies and final conclusions. Have you attempted to analyze the sensitivity of subsampling different proportions of zeros in your final assessments? For example, changing \( p \) and \( q \) in Section 3.1 that you can easily handle computationally?
-  When defining the Bayesian hierarchical models in Equations (1) and (4), I found that using the notation $\theta$ and $\theta^{size}$ at the data level is confusing and not in-line with traditional Bayesian hierarchical models. According to your notation, the top layer (data level) also depends on hyperparameters that are not related to the data level. For example, conditional on the log-intensity in (1), the distribution of $N_i$ does not depend on the hyperparameters $\theta$. Similarly, in equation (4), hyperparameters $\theta^{\text{size}}$ only appear at the process level,  and $\phi$ is the only hyperparmeter that is related to the data level. I have similar comments regarding equation (5) as well where $\sigma$ is the hyperparameter related to data level and not $ \theta^{year}$. I will try to remove these thetas from the data level to avoid any confusion for readers.
- I'm curious about the estimated spatial random effects in Figure 4; they appear very small, almost close to zero. You should provide some uncertainty estimates to determine if these effects are truly significant, and whether spatial effects are needed in the size model. Furthermore, there seems to be a dip in the estimated FA and FWI for larger FA and FWIs, which appears counterintuitive. Do you have any insights into why this might be the case?

## Minor comments
-  In Eqn (1), "size" is used as the superscript of $\theta^{YEAR}$ which might be a typo?
- In Figure 1, you might want to complete the sentence after "over"?
- In line 204, $B^{\text{eff}}$ should be in bold.
- In line 514 references, capitalize "Safran" to "SAFRAN."

**Strengths And Weaknesses:**

I had some difficulty understanding the novelty of the paper compared to Koh et al. (2023) and Pimout et al. (2021). I believe that the model proposed herein could be considered a sub-model of Koh et al. (2023). The one novelty that I could identify was in simulating future wildfires while combining the posteriors of model parameters with climate model output. This is a great idea, though I feel that this can be further improved, especially in the way the predictor variables FWI and FR are created for future projections. I'm wondering whether FWI and FR cannot be predicted or calculated based on reanalysis data or using available historical data, and then using these predicted FWI and FR when simulating future wildfires? Please disregard if this involves too many technicalities. However, assuming FWI and FR surfaces remain constant above historical ranges while projecting into the future might not be the most accurate approach, especially when using scenarios like RCP8.5. These clarifications should be clearly stated in the manuscript and/or in the discussion section if your goal is to do it as future research.

Another concern of mine is related to jointly modeling counts and size processes by the use of sharing random effects. You did mention the possibility of sharing some of the spatial random effects between the occurrence model and the size model, similar to Koh et al. (2023). While computational cost is one consideration, I believe you might see improved performance due to strong dependencies between the two processes and enhanced uncertainty estimations. This is something I will recommend to worth give a try to a datasets that you can handle computationally.  Again, this is the trade-off between model flexibility and computational costs.

Furthermore, the model proposed here includes spatial and temporal random effects in an additive fashion, hence lacking space-time interaction. I think this effect is evident from Figure 8, where the spatial pattern of estimated counts and burnt areas for future projections appears very similar to those in historical periods. I believe this could be an artifact of using a model that lacks space-time interactions. For instance, the models proposed herein account for temporal dependence, which is independent a priori from spatial effects. A proper spatiotemporal model would incorporate non-separable space-time structures to effectively capture space-time interactions. Therefore, I will be careful while drawing any such conclusions in future projections with models without space-time interaction. It's also possible that the observed similarity in spatial patterns is influenced by the local climate and specific to the study region.

---

### Note · Reviewer_iH5f · 2024-04-12

**Audience:**

Yes

**Claims And Evidence:**

Yes

**Decision Recommendation:**

Leaning Accept

---

### Note · Reviewer_hLqv · 2024-04-27

**Comment:**

I recommend acceptance after my comments have been addressed

**Audience:**

Yes

**Claims And Evidence:**

Yes

**Decision Recommendation:**

Leaning Accept

---

### Note · Reviewer_DqP1 · 2024-05-03

**Comment:**

The article is well-written and organized, providing an enjoyable reading experience. Moreover, it seems to align with the scope of Computo journals, and the reproducibility of the code is reassuring. However, I still believe there are several aspects highlighted in the reviewers' comments that could potentially enhance the novelty and quality of the paper. My decision `Leaning Accept' is contingent upon the authors addressing these necessary comments.

**Audience:**

Yes

**Claims And Evidence:**

Yes

**Decision Recommendation:**

Leaning Accept

---

### Decision · Action_Editor_QAfA · 2024-04-29

**Recommendation:** Accept with minor revision

**Comment:**

I suggest clarifying the novelty of the approach in comparison to Koh et al. (2023) and elucidating the modeling choices. Furthermore, it appears beneficial to provide deeper insights into the potential applications of the approach across various fields.

**Audience:**

The paper holds significance for Computo's audience, primarily owing to its pragmatic utilization of INLA SPDE within the framework of Marked Spatial Point Processes. Furthermore, the application is of considerable interest, given the criticality of wildfires and the paper's facilitation of projections under various climate change scenarios. The deployment of the proposed model is  well-justified, and meticulously articulated. Nevertheless, I recommend augmenting the discourse concerning the variety of  potential applications of the proposed approach, both in the introduction and conclusion sections. Such enhancement would better elucidate the methodology's expansive scope for prospective readers.

**Claims And Evidence:**

Dear Juliette Legrand,

The reviewers agree that the proposed work is of high quality, noting it's well-written and easy to follow. They suggest clarifying the modeling choices and improving how the innovations presented in the paper relate to the work of Koh et al. (2023)

---

> ### Decision · Editors_In_Chief · 2024-06-21
>
> I approve the AE's decision.